https://doi.org/10.1038/s41467-022-30407-3　**OPEN**

# Structural basis for inhibition of the Cation-chloride cotransporter NKCC1 by the diuretic drug bumetanide

Yongxiang Zhao[1], Kasturi Roy [2], Pietro Vidossich[3], Laura Cancedda [3], Marco De Vivo [3], Biff Forbush [2✉] & Erhu Cao [1✉]

Cation-chloride cotransporters (CCCs) NKCC1 and NKCC2 catalyze electroneutral symport of 1 Na⁺, 1 K⁺, and 2 Cl⁻ across cell membranes. NKCC1 mediates trans-epithelial Cl⁻ secretion and regulates excitability of some neurons and NKCC2 is critical to renal salt reabsorption. Both transporters are inhibited by the so-called loop diuretics including bumetanide, and these drugs are a mainstay for treating edema and hypertension. Here, our single-particle electron cryo-microscopy structures supported by functional studies reveal an outward-facing conformation of NKCC1, showing bumetanide wedged into a pocket in the extracellular ion translocation pathway. Based on these and the previously published inward-facing structures, we define the translocation pathway and the conformational changes necessary for ion translocation. We also identify an NKCC1 dimer with separated transmembrane domains and extensive transmembrane and C-terminal domain interactions. We further define an N-terminal phosphoregulatory domain that interacts with the C-terminal domain, suggesting a mechanism whereby (de)phosphorylation regulates NKCC1 by tuning the strength of this domain association.

[1] Department of Biochemistry, University of Utah School of Medicine, Salt Lake City, UT 84112-5650, USA. [2] Department of Cellular and Molecular Physiology, Yale University School of Medicine, New Haven, CT, USA. [3] Istituto Italiano di Tecnologia, Via Morego 30, 16163 Genova, Italy. ✉email: biff.forbush@yale.edu; Erhu.Cao@biochem.utah.edu

Loop diuretics such as bumetanide and furosemide, which are marketed as Bumex and Lasix, respectively, are cornerstones of clinical management of edema and hypertension[1–5], with more than 30 million prescriptions each year in the United States. Bumetanide inhibits the renal-specific cation-chloride cotransporter (CCC) NKCC2, thereby reducing salt and water retention by the kidney to exert diuretic and anti-hypertensive effects[6,7]. Bumetanide (and other existing loop diuretics) also antagonize a more widely distributed paralogous transporter NKCC1, which accounts for their side-effect of ototoxicity[8,9]. This side effect is consistent with the genetic studies that NKCC1$^{-/-}$ animals are profoundly deaf because NKCC1 contributes to maintenance of unusually high extracellular [K$^+$] of endolymph in inner ear[10,11].

An antagonistic action of bumetanide on neuronal NKCC1 may also produce beneficial effects in the treatment of various brain disorders and psychiatric conditions as demonstrated in recent off-label studies[12–17]. NKCC1 may function as a major Cl$^-$ importer in neurons and thus contributes to maintenance of an outward-directed Cl$^-$ electrochemical gradient in immature neurons and some peripheral sensory neurons in adults[18–20]. Consequently, inhibitory neurotransmitters (e.g., $\gamma$-aminobutyric acid; GABA) stimulate Cl$^-$ efflux via pentameric ligand-gated Cl$^-$ channels, resulting in depolarization of these neurons[21,22]. Bumetanide may restore GABA inhibition in patients with neurological disorders by blocking NKCC1-mediated Cl$^-$ accumulation in neurons. However, bumetanide barely crosses the blood brain barrier to reach neuronal NKCC1 target and has the diuretic liability associated with off-target inhibition of the closely related NKCC2 transporter[23].

We and others have recently reported a number of single-particle electron cryo-microscopy (cryo-EM) structures of CCCs[24–31], thereby defining their dimeric architectures and providing valuable insights into ion binding mechanisms. These structures reveal that the CCC transport core comprises two five-helix bundle inverted repeats (TM1-5 and TM6-10) that are related by a pseudo two-fold symmetry, with the two remaining TM helices (TM11 and TM12) mediating dimerization of subunits within the bilayer. Cytoplasmic and extracellular domains also contribute to inter-subunit association of CCCs. The TM1 and TM6 helices lie at the center of ion translocation path. Both helices break α-helical geometry roughly in the middle of the lipid bilayer, and ion binding sites are organized around these discontinuous hinge regions between half helices. A significant limitation of the published CCC structures, however, is that they were all captured in an inward-open state and therefore do not reveal the postulated outward-open and occluded states that CCCs must sample to shuttle ions across membrane.

CCCs are regulated by the WNKs-SPAK kinase signaling cascade through phosphorylation of key cytoplasmic serine/ threonine residues[32–34]. Phosphorylation of a conserved N-terminal segment in Na$^+$-Cl$^-$ (NCC) and NKCCs enhances ion transport, whereas dephosphorylation by PP1 or other phosphatases inhibits these transporters. Importantly, mutations in WNK1, WNK4, or their upstream ubiquitin E3 ligase degrader CUL3/KLHL3 all enhance transport activity of NCC/NKCC2 and cause salt-sensitive hypertensive Gordon syndrome[35,36]. However, because the N-terminal phosphoregulatory domain of NKCC1 has not been defined in structural terms in previous studies[24,30,31], it remains unclear whether/how (de)phosphorylation of the N-terminal segment triggers conformational changes that is then transmitted to impact the membrane-embedded ion translocation pathway.

Here we determine cryo-EM structures of human NKCC1 bound with bumetanide, providing an initial blueprint to enable the development of NKCC1- and NKCC2-specific inhibitors for the treatment of neurological disorders versus edema/hypertension. Our NKCC1/bumetanide structure intriguingly adopts a dimeric architecture with two interdigitating C-terminal domains in close contact with two disengaged transmembrane units, revealing an unappreciated plasticity in NKCC1 inter-subunit interfaces. We also discover two coupled domain interfaces wherein an intracellular N-terminal segment interacts with the C-terminal domain which also associates with the transmembrane domain that houses the ion translocation pathways. This N-terminal segment represents the elusive phosphoregulatory domain as it bears key phosphoacceptor residues, thus hinting at a mechanism whereby kinases and phosphatases may regulate NKCC1 transport activity by oppositely tuning the strength of association between its cytosolic N- and C-terminal domains.

## Results

**Cryo-EM analyses of human NKCC1 bound with bumetanide.** To obtain a biochemically stable NKCC1/bumetanide complex suitable for structural studies, we relied on the knowledge that bumetanide binds only to the active transporter[37,38]. We thus used a full-length human NKCC1 construct that bears A492E and L671C mutations, each of which shows enhanced relative transport activity at elevated intracellular Cl$^-$ concentrations ([Cl$^-$]$_i$) which normally bring about the inactive state; the A671C mutant actually shows considerable constitutive activation (Supplementary Fig. 1). We additionally introduced a functionally neutral K289N substitution that we previously used to increase expression of NKCC1[30], resulting in a construct that we hereafter referred to as NKCC1$_{iii}$. We also generated a full-length human NKCC1$_{ii}$ construct that harbors K289N and A492E mutations and found that NKCC1$_{ii}$ exhibited ~70% of transport activity relative to the wildtype transporter (Supplementary Fig. 2). NKCC1$_{iii}$ exhibited a transport activity below our detection limit in the insect cell Cl$^-$ influx assay, but mediated a small but significant bumetanide-sensitive Cl$^-$ influx in HEK293 cells, ~5% of the wildtype activity (Supplementary Figs. 2 and 3). When compared to the wildtype transporter, NKCC1$_{iii}$ exhibited similar ion requirements for transport as indicated by an analysis of the K$^+$-dependence of Cl$^-$ influx, but showed ~4.5-fold higher affinity for bumetanide, a behavior consistent with stabilization of outward-open conformations. Förster resonance energy transfer (FRET) experiments indicated that the C-terminal domain is similarly involved in regulation of NKCC$_{iii}$ activation. However, the FRET decrease we recorded upon calyculin A administration, a phosphatase inhibitor that promotes phosphorylation/activation of NKCC1, was only about half the magnitude seen in the wildtype construct similarly tagged with the FRET pair (Supplementary Fig. 4).

The NKCC1$_{ii}$ and NKCC1$_{iii}$ constructs enabled determination of structures of NKCC1$_{iii}$/bumetanide, NKCC1$_{ii}$/bumetanide, and NKCC1$_{iii}$/apo at 2.9 Å, 3.6 Å, and 3.3 Å resolution (Fig. 1a, b and Supplementary Figs. 5–11), respectively. Of note, the NKCC1$_{iii}$/ bumetanide structure defines a conformation of NKCC1 dimer in which the two interdigitating C-terminal domains dictate dimeric assembly, while the two transmembrane units are fully separated within the lipid bilayer. NKCC1$_{ii}$/bumetanide and NKCC1$_{iii}$/apo resemble reported NKCC1 dimers where two transmembrane units associate via an inverted V-shaped TM11-turn-TM12 structure[30,31]. Bumetanide adopted an identical binding pose and arrested NKCC1 in a similar outward-open state in both NKCC1$_{iii}$/bumetanide and NKCC1$_{ii}$/bumetanide structures (Supplementary Fig. 12a). We will hereafter focus our discussion of bumetanide action on NKCC1 using the former higher resolution structure unless stated otherwise.

**Bumetanide binds to an extracellular vestibule of NKCC1.** Structures of CCCs so far all assume an inward-open state[24–31], except for our recent outward-open K$^+$-Cl$^-$ cotransporter 1

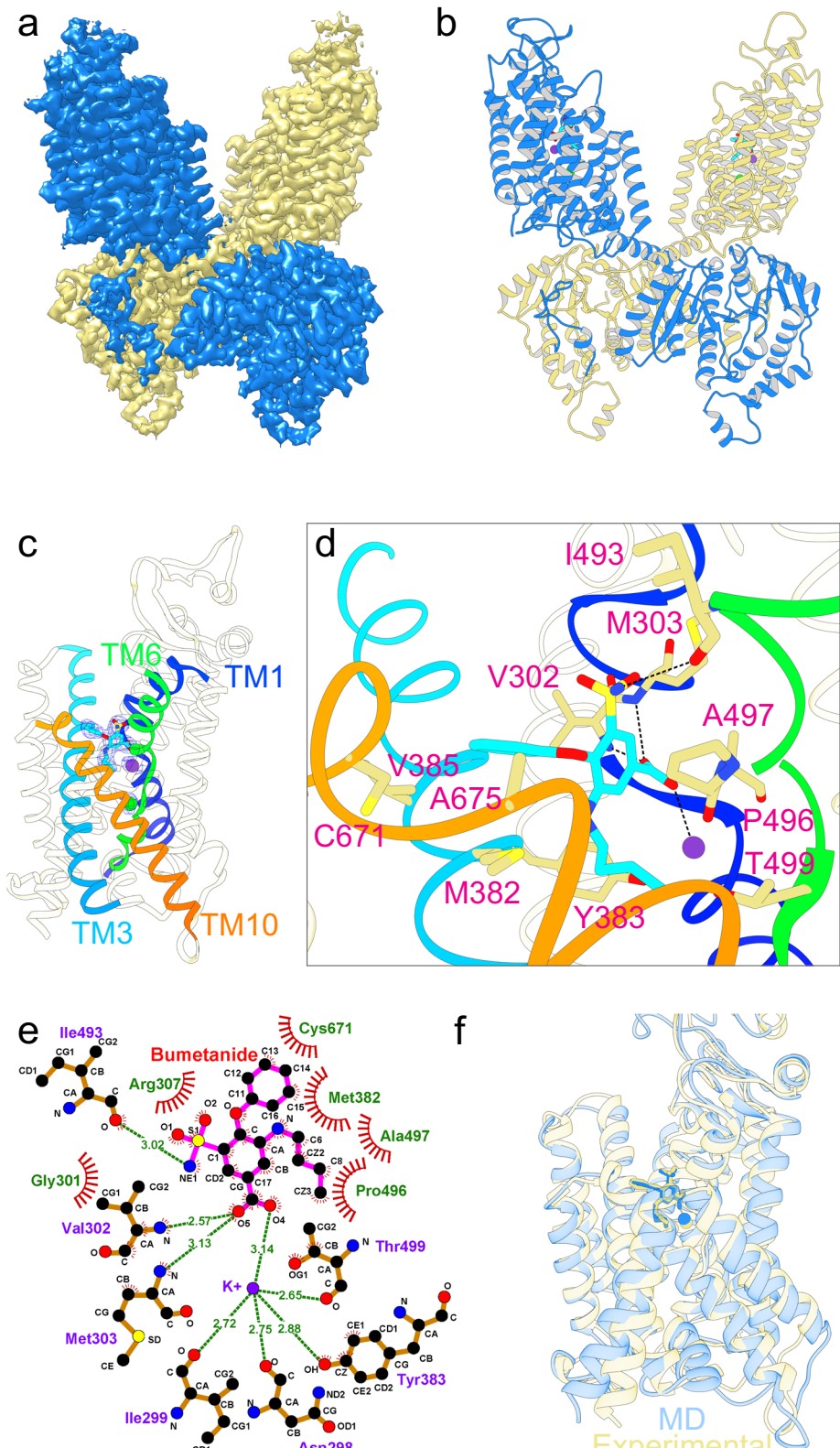

**Fig. 1 Bumetanide binds to the extracellular entryway of NKCC1.** Overall architecture of human NKCC1 bound with bumetanide shown in map (**a**) and ribbon diagram (**b**). **c** A single NKCC1 transmembrane domain highlights bumetanide (stick), K+ (purple sphere), and Cl− (green sphere). Blue meshes are experimental cryo-EM densities. **d** Bumetanide binding pocket highlights key interacting residues along the extracellular ion pathway. **e** A 2D representation of bumetanide interactions with residues along the extracellular ion permeation path. Each eyelash indicates a hydrophobic interaction. **f** Superimposition of experimental NKCC1/bumetanide structure and a MD simulation model taken at 1000 ns shows almost identical binding pose of bumetanide. The K+ ions shown as spheres also maintain its position during simulations.

(KCC1) structure in which the VU0463271 inhibitor wedges into and plugs an extracellular vestibule that would otherwise allow for unobstructed access of ions to their central binding sites[39]. Bumetanide analogously fits into an extracellular pocket formed by TM helices 1b, 6a, 3, and 10 (Fig. 1c), highlighting the extracellular ion entryway as a hotspot for pharmacological inhibition of both $Na^+$-dependent and $Na^+$-independent clades of CCCs. Bumetanide adopts a cross-shaped configuration, with the benzene ring residing at the center and its four roughly orthogonally arranged substituting groups making a number of polar and hydrophobic contacts with residues along the extracellular ion translocation path (Fig. 1d, e and Supplementary Fig. 13a). One $K^+$ and one tentatively assigned $Cl^-$ ion at $Cl^-_{site2}$ are co-occluded with bumetanide - but no $Cl^-$ ion was observed at $Cl^-_{site1}$ because it is occupied by the carboxyl group of bumetanide (Supplementary Fig. 12b; also see below). As with the reported inward-open zebrafish NKCC1 structure[24], our outward-open human $NKCC1_{iii}$/bumetanide map also did not unambiguously locate $Na^+$. These structural findings are consistent with the result that $K^+$ and $Cl^-$ (and $Na^+$) are required for bumetanide binding[40] and that both $K^+$ and $Cl^-$ are occluded within NKCC1 as long as bumetanide is bound (Supplementary Fig. 14). Our structure also explains the known biphasic effect of $Cl^-$ on [$^3$H]bumetanide binding to NKCCs such that bumetanide binding is initially enhanced by raising [$Cl^-$] up to ~ 30 mM, but then progressively declines with further increasing [$Cl^-$][38,40].

At the mouth of the extracellular ion entryway, bumetanide's sulfamyl group interacts with Ile493 of TM6a, possibly explaining why non-sulfonamide loop diuretics (e.g., ethacrynic acid) are significantly less potent than bumetanide (Fig. 1d, e and Supplementary Fig. 13a)[41]. Beneath this top layer of interaction, the phenoxyl and carboxyl groups project toward opposite sides of the ion translocation path and interact with residues residing on TM1b, TM3, and TM10. Here on one side, the phenoxyl group packs against Val385 residing on TM3, and Leu671 (Cys671 in the $NKCC1_{iii}$ construct) and Ala675 located at TM10. In furosemide, this phenoxyl group is replaced by a chlorine atom, likely explaining why it is ~ 50-fold less potent in inhibiting NKCCs than bumetanide (Supplementary Fig. 13a)[42]. Indeed, our 3.4 Å $NKCC1_{ii}$/furosemide structure, which almost superimposes with the outward-open NKCC1/bumetanide structures, confirmed a general binding site for loop diuretics in NKCC1, but the precise pose for furosemide could not be determined based on the current map (Supplementary Fig. 15). On the other side, the carboxyl group forms hydrogen bonds with Val302 and Met303 located on TM1b, and Ala497 residing on TM6a; it also replaces a $Cl^-$ ion in the $Cl^-_{site1}$ and establishes ionic interaction with the $K^+$ ion. Deeper toward the central ion binding sites, the butylamino group establishes hydrophobic interactions with Met382 and Tyr383 located on TM3, and Pro496 and Thr499 residing on TM6a. This explains why substitution of Met382 to tryptophan abolishes inhibition of NKCC1 by bumetanide[43].

We performed molecular dynamic (MD) simulations to validate the binding pose of bumetanide seen in our NKCC1/bumetanide structures. We found that bumetanide maintains the experimental pose during 1000 ns simulations with a root mean square deviation (RMSD) of ~1 Å (Fig. 1f and Supplementary Fig. 13b). Taken together, we showed that bumetanide, and probably other chemically-related loop diuretics as well, inhibit NKCC1 by selectively plugging the extracellular ion entry pathway created in an outward-open state and sterically hinder the transporter from isomerization into other transport states.

**An N-terminal phosphoregulatory segment associates with the C-terminal domain**. NKCC1, NKCC2, and the related $Na^+$-$Cl^-$ cotransporter (NCC) are activated by the WNK-SPAK kinase cascade that phosphorylates several key threonine/serine residues located in a conserved N-terminal segment of these transporters[32–34]. In human NKCC1, Thr217 represents a major phosphoacceptor site indispensable for transporter activation in response to cell volume shrinkage or stimulation by secretagogues such as isoproterenol[44,45]; Thr230 may also contribute to phosphoregulation. Our $NKCC1_{iii}$/bumetanide map structurally defined the elusive N-terminal phosphoregulatory domain composed of residues from Asn216 to Leu250 which is seen to interact with a swapped C-terminal domain of a second NKCC1 subunit (Fig. 2a and Supplementary Fig. 11). Such an N- and C-terminal domain interaction has been previously hypothesized to underlie phosphoregulation of CCCs[31]. The N- and C-terminal domain association is stabilized via extensive hydrophilic contacts that involve three layers of residues (Fig. 2b). In the top layer located ~20 Å beneath the inner membrane, the N-terminal residues His226, Tyr227, and Arg240 establish a network of polar interactions with residues Asp1031, Gly1033, and Gly1034 residing in a loop buried within the C-terminal domain and His1204, Gln1205, and Ser1206 located in the extreme C-terminal tail. In the middle layer, the N-terminal residue Asp219 forms a salt bridge with the C-terminal residue Arg1201. Disruption of these hydrophilic interactions, as well as several buried hydrophobic residues within the N-terminal segment, significantly reduced NKCC1 transport activity; of particular note is complete loss of activity with mutation of L243 to alanine (Fig. 2c and Supplementary Figs. 16 and 17). At the bottom layer, the phosphoacceptor Thr217 appears to form a hydrogen bond with the C-terminal Arg1174. This constitutive interaction likely explains why substitution of this threonine to alanine (or serine) abolishes both human and shark NKCC1 transport activity (Supplementary Fig. 18)[44]. We also examined a set of mutants in the N-terminal regulatory domain that we hope would strengthen association between N- and C-terminal domains (Supplementary Fig. 18a). All these mutants, including T217E and T230E phosphomimic, did not show increased transport activity (Supplementary Fig. 18b, c). We also attempted to identify a second mutation in the N-terminal segment that can rescue the T217A loss-of-function phenotype to no vail (Supplementary Fig. 18d, e). It remains uncertain how conjugation of a negatively charged phosphate group to Thr217 (and Thr230) by kinases (e.g., SPAK) could lead to an altered interaction between the N- and C-terminal domains which could then favorably interact with the transmembrane core or alter dimeric architecture to enhance ion transport activity of NKCC1.

**Bumetanide arrests NKCC1 in an outward-open conformation**. Although a ligand-free outward-open CCC structure is yet to be determined, bumetanide may selectively bind to and stabilize an existing physiologically relevant outward-open state as NKCC1 proceeds along its transport cycle. Opening of an extracellular vestibule and accommodation of bumetanide is brought about through a combination of subtle side chain movements of extracellular gating residues and rigid body outward displacements of TM3, TM9, and TM10 helices (Fig. 3a–c, Supplementary Fig. 19a, and Supplementary Movie 1). First, the salt bridge formed between Arg307 (TM1b) and Glu389 (TM3) breaks caused by movement, and consequently, misalignment/separation of side chains of these two involving residues (Fig. 3c). This salt bridge is crucial for closure of the extracellular gate in all inward-open NKCC1 structures[24,30,31]. Second, bumetanide also directly engages with the extracellular portion of an inverted V-shaped TM9-turn-TM10 structure, leading to a downward movement of the entire TM9 helix by about a half helix turn (~3.2 Å) and an

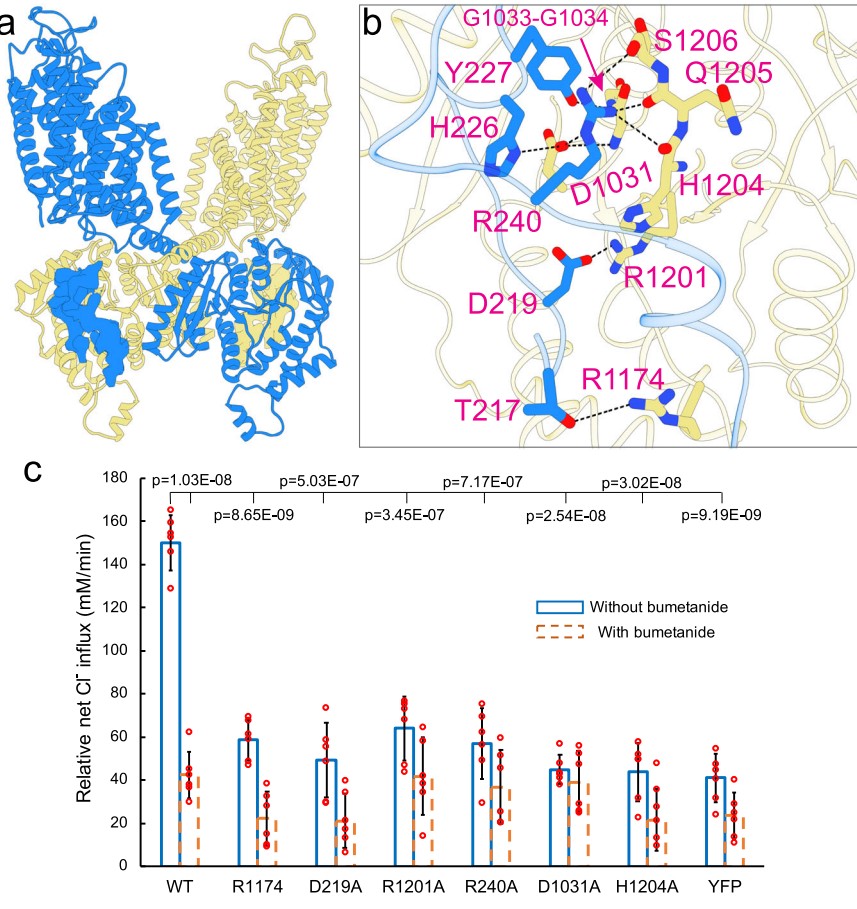

**Fig. 2 An N-terminal phosphoregulatory segment associates with the C-terminal domain. a** An N-terminal segment shown as density map interacts with a swapped C-terminal domain shown as ribbon. **b** An enlarged view highlights residues involved in association of the N- and C-terminal domains. Polar contacts are indicated as dashed lines. **c** Cl$^-$ transport rates in insect cells of wildtype NKCC1 and mutants designed to disrupt the N- and C-terminal domain interface. Each circle represents one kinetic measurement of a single sample in a 96 well plate. Unpaired one-tailed Student's $t$ tests are used for statistical analyses ($n = 6$; data are presented as mean values ± SD).

outward displacement of the extracellular half of the TM10 helix by ~3.5 Å away from the extracellular vestibule.

Supporting the accessibility of this vestibule to the extracellular side, MD simulations showed that the outward-open state trapped by bumetanide remains stable for the duration of 500 ns after bumetanide is removed from the structure (Supplementary Fig. 13c). In addition, we showed that cysteine substitution of residues lining the extracellular vestibule (M382C, P496C, and S679C) confers sensitivity to inhibition by the cysteine reactive MTSET (Supplementary Fig. 20), extending previous reports of the accessibility of vestibule cysteines[43,46]. Our structural findings also agree with previous crosslinking and solvent accessibility studies of NKCC1 that indicate significant movement of TM10 during active ion transport[47].

Congruent with the alternate access model, opening of an extracellular vestibule in NKCC1 leads to concomitant closure of a cytoplasmic exit. In the outward-open NKCC1$_{iii}$ and NKCC1$_{ii}$/bumetanide structures, the cytoplasmic vestibule is occluded by inward movements of TM4, TM5, TM8, a short helix within the intracellular loop 1 (ICL1), and the ICL4 that connects the TM8 and TM9 helices (Fig. 3b, Supplementary Fig. 19a, and Supplementary Movie 1). In particular, the TM4 helix, as with the TM9 helix, moves downward toward the inner membrane by about a half helix turn, possible because these two anti-parallel helices are coupled through extensive hydrophobic interactions. This causes an inward movement of the intracellular portion of the V-shaped TM4-turn-TM5 structure, which, together with

inward displacement of TM8 and the short ICL1 helix, occlude the cytoplasmic exit. Moreover, Met428 located on TM5 also impedes ion flow by projecting its hydrophobic side chain into the cytoplasmic vestibule (Fig. 3c).

Newly established gating interactions at the cytoplasmic side further stabilize the outward-open conformation (Fig. 3c, Supplementary Fig. 19a, and Supplementary Movie 1). In particular, Asp510 and Lys624, residing at the intracellular ends of TM6b and TM8 helices, respectively, form a salt bridge at the mouth of the cytoplasmic exit. Above the Asp510-Lys624 pair further toward the central ion binding sites, Arg294 (TM1a) and Glu431 (TM5) establish another ionic interaction. Mutations of these gating residues Asp510, Lys624, and Arg294 all significantly decrease ion transport rate[30,31]. We note that gating interactions at the extracellular entryway (Arg307-Glu389 in the inward-open state) and the cytoplasmic exit (Asp510-Lys624 in outward-open state), as well as reciprocal movements of the TM4-turn-TM5 and TM9-turn-TM10 structures with respect to the ion permeation path, are related by a pseudo two-fold symmetry. These findings reinforce the concept that the intrinsic symmetry within the LeuT-fold transporters underpins the alternate access mechanism of substrate transport[48,49].

Superimposition of NKCC1/bumetanide and our recent KCC1/VU0463271 structures[39] revealed that NKCC1 and KCC1 adopts a similar outward-open conformation stabilized by inhibitors that bind to an analogous receptor site at the extracellular ion entryway (Fig. 3d). In both NKCC1 and KCC1, the bundle helices

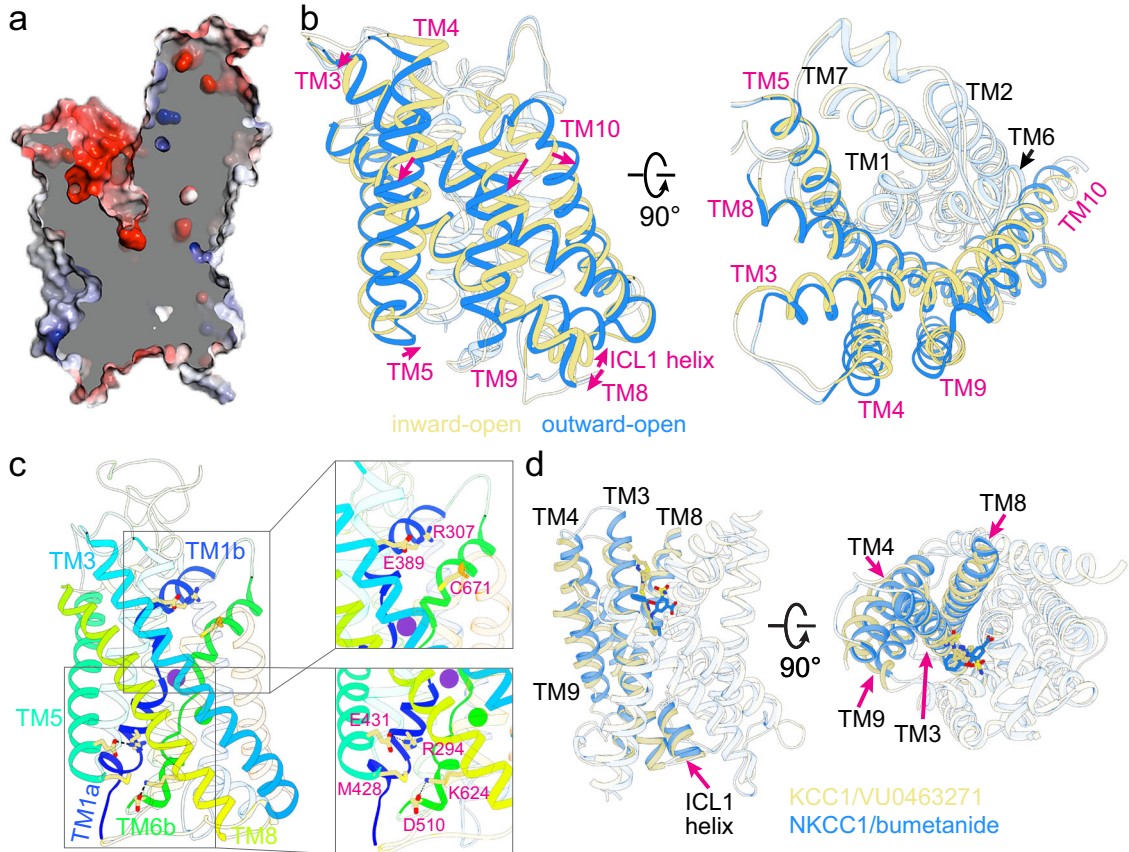

**Fig. 3 Conformational changes upon isomerization between outward-open and inward-open states in NKCC1. a** A "slab" view of the electrostatic potential map of the outward-open NKCC1iii/bumetanide structure, highlighting an extracellular vestibule if bumetanide is removed. **b** Superimposition of inward- (khaki) and outward-open (dodger blue) NKCC1 structures. Movements of helices are indicated as red arrows. **c** Left, gating interactions at the extracellular ion entryway and cytoplasmic exit in the outward-open NKCC1 structure. Right, two zoomed views highlight the broken salt bridge (R307-E389) seen in inward-open structures, and a newly formed ionic interaction (D510-K624) that closes the intracellular exit. **d** Superimposition of outward-open structures of NKCC1/bumetanide and KCC1/VU0463271 highlights a similar outward-open state and an analogous receptor site for inhibitors at the extracellular vestibule.

(TM1, TM6, TM2, and TM7) remain largely static, whereas the scaffolding helices (TM3, TM8, TM4 and TM9) undergo displacements (albeit to various extents) as these transporters transition between inward- and outward-open states. These findings suggest that isomerization between inward- and outward-open states in CCCs do not necessitate hinge-bending motions around the discontinuous regions of TM1 and TM6 helices as seen in LeuT and proposed in other related amino acid-polyamine-organocation (APC) transporters (Fig. 3d, Supplementary Fig. 19b, and Supplementary Movie 2)[48–50].

**NKCC1 dimer defines a transmembrane and C-terminal domain interface.** In our NKCC1iii/bumetanide structure, the two transmembrane units are separated by approximately 35 Å within the lipid bilayer, whereas the two C-terminal domains interdigitate to hold two subunits together with a buried surface area of ~ 5819 Å² (Fig. 4a). Most importantly, the two disengaged transmembrane domains are now well positioned to associate with the C-terminal domains via extensive interactions not seen in the zebrafish NKCC1 structure (Fig. 4b). In particular, residues located on ICL1 and ICL5 establish a number of polar contacts with a swapped C-terminal domain of a second NKCC1 subunit. For instance, the ICL1 residues Arg348 and Tyr354 interact with the extreme C-terminus residues Tyr1211 and Ser1212. Moreover, the ICL5 residue Arg705 interacts with C-terminal residue Ser1209. In addition, the C-terminal residues Lys785 and

Arg1080 form hydrogen bonds with residues located on ICL1 and ICL5. The C-terminal and transmembrane domain interface may regulate NKCC1 activity as weakening of domain association by site-directed mutagenesis reduces rates of ion transport (Fig. 4c and Supplementary Fig. 21). Of note, the conserved extreme C-terminal tail (Arg1201-Ser1212) interacts with both the N-terminal phosphoregulatory segment and ICL1 (Fig. 2b and Fig. 4a), hinting at potential allosteric communications among these structural elements that may be subjected to regulation by kinases and phosphatases.

Separation of the two transmembrane units seen in our NKCC1iii/bumetanide structure drastically differs from NKCC1iii/apo, NKCC1ii/bumetanide, and previously reported NKCC1 structures in which an inverted V-shaped TM11-turn-TM12 structure mediates inter-subunit association within the lipid bilayer (Fig. 5a and Supplementary Figs. 8–10). In these transmembrane associated NKCC1 dimers, the C-terminal domains were only well-resolved in the zebrafish[24], but not human NKCC1 structures (Supplementary Figs. 8–10)[26,30], possibly because the C-terminal domain dimer tends to loosely associate with and thus could assume a range of orientations with respect to the transmembrane domains. Whether all these NKCC1 dimeric forms exist in cells remain to be determined, but rupture of the transmembrane interface seen in our NKCC1iii/bumetanide structure may be possible as this interface only buries a surface area of ~500 Å² as compared to ~5819 Å² buried between the two C-terminal domains. The separation of the

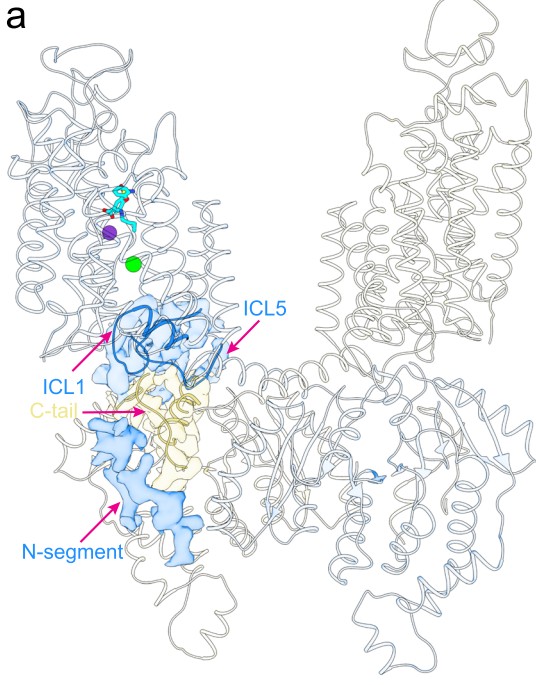

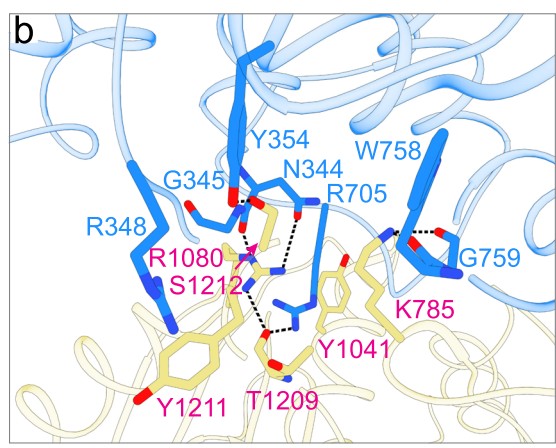

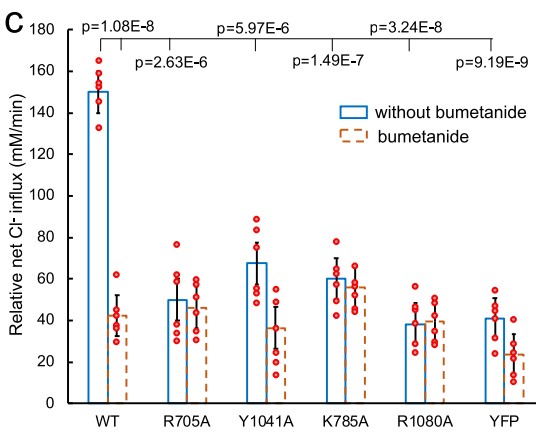

**Fig. 4 A transmembrane and C-terminal domain interface regulates NKCC1 activity. a** A NKCC1 dimer structure highlights coupling among the N-terminal phosphoregulatory segment, extreme C-terminal tail, and ICL1. **b** The interface between the transmembrane (dodger blue) and a swapped C-terminal (khaki) domains is observed only in the herein described form of NKCC1iii/bumetanide dimer. Polar interactions are indicated as dashed lines. **c** Cl⁻ transport rates in insect cells of wildtype human NKCC1 and mutants designed to disrupt the transmembrane and C-terminal domain interface. Each circle represents one kinetic measurement of a single sample in a 96 well plate. Unpaired one-tailed Student's *t* tests are used for statistical analyses ($n = 6$; data are presented as mean values ± SD).

outward-open NKCC1iii/bumetanide, inward-open zebrafish NKCC1, and alphafold-predicted human NKCC1 structures to pinpoint structural elements whose conformational changes may trigger conversion between these two dimeric architectures. Superimposition of experimental dimeric human and zebrafish NKCC1 structures showed that their C-terminal domain dimers nicely superimpose. However, a flexible loop, which connects the TM12 helix and the C-terminal domain scissor helix, adopts different conformations in these NKCC1 structures, giving rise to drastically distinct placement of their transmembrane domains within the lipid bilayer (Fig. 5a). In our NKCC1iii/bumetanide structure, the loop between the TM12 and scissor helices, the ICL1 helix, and the cytoplasmic ends of TM3 and TM8 are coupled through hydrogen bond, salt bridge, and cation-π interactions (Fig. 5b). Some of these interactions are absent in zebrafish and alphafold-predicted NKCC1 structures, possibly explaining why this loop can assume distinct conformations.

## Discussion

Here we reveal the mechanism of action of bumetanide on NKCC1, five decades after it was first developed for the clinical management of fluid overload and hypertension[51,52]. Inhibition of NKCC1 by bumetanide and other loop diuretics is a double-edged sword in clinical medicine. On the one hand, antagonizing neuronal NKCC1 represents an emerging attractive strategy for the treatment of a wide range of brain disorders such as seizure and autism[14,53,54]. On the other hand, inhibition of NKCC1 also causes side effect of ototoxicity in the treatment of edema/hypertension[8,9]. We now show that bumetanide targets the extracellular ion translocation path that constitutes a set of residues almost invariant in human NKCC1 and NKCC2, explaining why bumetanide (and other loop diuretics) cannot discriminate these two closely related paralogues (Supplementary Fig. 22). Our studies suggest that isoform-specific inhibitors of NKCCs could be better crafted by interfering with their respective regulatory mechanisms which must have adapted to fulfill their divergent roles. One such enticing strategy would be to develop small molecules (or biologics) that can modulate the strength of association between their N-terminal phosphoregulatory segment and C-terminal domain.

The structures and functional studies of NKCC1 presented here allow us to appreciate a membrane-embedded pathway for the cotransported 1 Na⁺, 1 K⁺, and 2 Cl⁻ ions from the extra-cellular to intracellular medium. According to the so-called "glide symmetry" model[55], an outward-open NKCCs transporter first binds a Na⁺, then a Cl⁻, followed by a K⁺ and a second Cl⁻. The transporter then switches to an inward-open conformation and ions are released in the same sequence as they bind externally. Using this model as a reference, our NKCC1/bumetanide structure likely captures the transporter in an outward-open state in which binding of the second Cl⁻ seems to be interrupted due to competition of bumetanide. Furthermore, although we observed very weak density at the Na⁺ site in our NKCC1iii/bumetanide

two transmembrane units is apparently not triggered by the outward-open conformation for two reasons: (1) the outward-open NKCC1ii/bumetanide structure also assumes a transmembrane associated arrangement; and (2) the TM11-turn-TM12 structure that mediates dimerization within the membrane assumes a similar orientation with respect to the TM1-TM10 helices regardless of whether NKCC1 adopts an inward- or outward-open state. We compared the

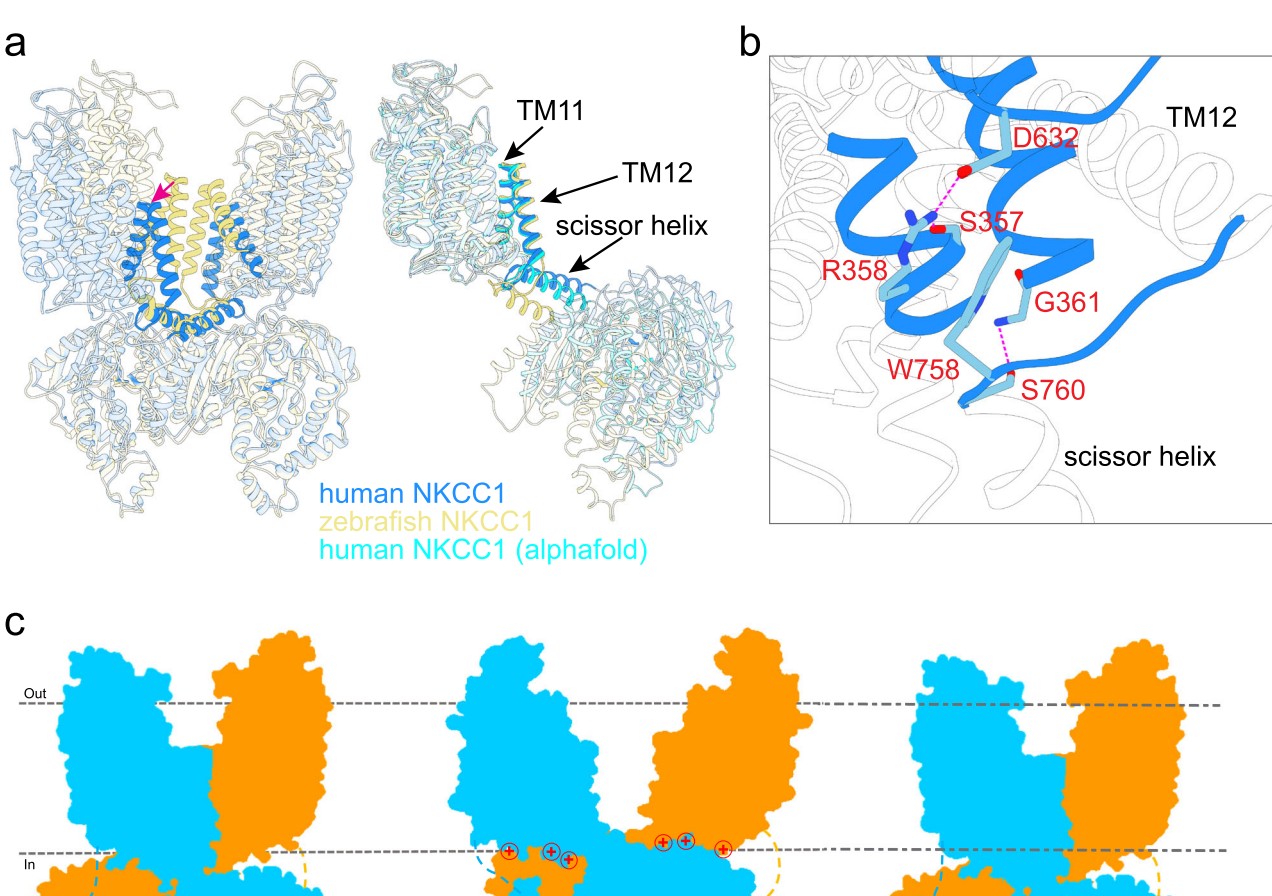

**Fig. 5 NKCC1_iii/bumetanide adopts a dimeric architecture. a** Left, superimposition of human and zebrafish NKCC1 dimers based on their C-terminal domains shows that the two transmembrane units in human NKCC1 separate and interact extensively with its C-terminal domains. Right, superimposition of a single human and zebrafish NKCC1 subunit based on their transmembrane domains shows displacement of the scissor helix and C-terminal domain as the loop that connects the TM12 and scissor helices assumes different conformations in these structures. **b** In the NKCC1_iii/bumetanide structure, the loop connecting the TM12 and scissor helices is stabilized by interactions with the transmembrane unit. Hydrophilic interactions are depicted as dashed lines. **c** A hypothetical model depicts NKCC1 regulation by (de)phosphorylation. Kinases and phosphatases possibly oppositely tune the strength of association between the cytosolic N- and C-terminal domains with weak and strong association indicted by faded and bright N-terminal regulatory segment, respectively. It remains uncertain whether phosphorylation leads to disengagement of the two transmembrane units or triggers dissociation of the two interdigitating C-terminal domains as indicated by the two oppositely directed red arrows.

map, we could not unambiguously locate the Na$^+$ ion. This suggests that bumetanide cannot completely prevent escape of Na$^+$ into the extracellular space, possibly because bumetanide does not directly interact with Na$^+$. In contrast, direct ionic interaction between bumetanide and K$^+$ prevents escape of K$^+$ from its binding site. We also identify small but essential conformational changes that must take place to flip ion binding site accessibility from one side to the other (Supplementary Fig. 19a and Movie 1). NKCCs have a relatively high transport turnover rate of 4000 s$^{-1}$ as estimated in duck red blood cells[38], orders of magnitude greater than LeuT; it seems likely that this is enabled by the relatively smaller conformational changes seen between outward- and inward-open NKCC1 structures compared to LeuT as previously proposed[43] (Supplementary Fig. 19b and Movie 2).

As we and others have demonstrated for KCC1 and other KCCs[25,27,39,56,57], NKCC1 can also assume drastically different

dimeric architectures whose roles in NKCC1 phosphoregulation await future explorations. We suspect that these different dimeric structures may capture NKCC1 in distinct activation states which could be regulated by the opposing actions of kinases and phosphatases (Fig. 5c). The herein described NKCC1 dimer features two previously unknown domain interfaces: one between the C-terminal domain and an N-terminal phosphoregulatory segment and another between the C-terminal and transmembrane domains. Together, these two herein defined domain interfaces define an allosteric network that could potentially transmit impacts of (de)phosphorylation of Thr217 (and Thr230) located at the N-terminal segment (Asn216-Leu250), via the intervening extreme C-terminal tail (Arg1201-Ser1212) and ICL1 structures, to ultimately influence the membrane-embedded ion translocation pathway (Fig. 4a). Weakening these interfaces via site-directed mutagenesis all reduces NKCC1 transport

activity possibly by disrupting allosteric communications among these domains. We hypothesize that kinases and phosphatases may regulate NKCC1 activity by catalyzing formation or rupture of these two domain interfaces (Fig. 5c). However, it remains unclear whether phosphorylation leads to disengagement of the transmembrane domains as seen in herein described NKCC1 dimer or dissociation of the C-terminal domains as previously proposed[31,58]. Our studies now set the stage to define conformational changes as NKCC1 alternates among outward-open, occluded, and inward-open states, and to understand how conversion among these dimeric forms may be regulated by (de) phosphorylation using the single-molecule Förster resonance energy transfer (smFRET) approach.

## Methods

**Expression and purification of human NKCC1.** The full-length human NKCC1 constructs were cloned into a modified pFastBac1 vector wherein expression of target genes is driven by the human cytomegalovirus (CMV) promotor. The NKCC1iii construct bears K289N, A492E, L671C mutations and the NKCC1ii construct harbors K289N and A492E mutations. A maltose binding protein (MBP) followed by a tobacco etch virus (TEV) protease cleavage site was fused immediately before the N-terminus of NKCC1. These constructs were expressed in HEK293S GnTI$^{-/-}$ cells (ATCC CRL-3022) using the BacMam system as described[59]. In brief, HEK293S GnTI$^{-/-}$ cells, grown in suspension in Freestyle 293 expression medium (Invitrogen, Carlsbad, CA) at 37 °C in an orbital shaker, were transduced with baculoviruses when cell density reached ~2 × 10$^6$/ml. 8–12 h post transduction, sodium butyrate was added to the culture to a final concentration of 5 mM to enhance protein expression; temperature was reduced to 30 °C. Cells were harvested 72 h post transduction and flash frozen in liquid nitrogen and stored at −80 °C until use. All protein purification steps were carried out at 4 °C unless stated otherwise. Membrane proteins were extracted for 1 h at room temperature in a buffer composed of (in mM) 50 HEPES (pH7.4), 75 KCl, 75 NaCl, 0.5 tris(2-carboxyethyl) phosphine (TCEP), 3 lauryl maltose neopentyl glycol (LMNG-3), and 0.6 cholesteryl hemisuccinate tris salt (CHS), 0.5 phenylmethylsulfonyl fluoride (PMSF), 5 μg/ml leupeptin, 1.4 μg/ml pepstatin A, 2 μg/ml aprotinin. The supernatant was collected after centrifugation at 18,000 rpm for 30 min and then incubated with amylose resin (New England BioLabs, Ipswich, MA) for 2 h. NKCC1 protein was eluted from amylose resin with buffer composed of (in mM) 20 HEPES (pH7.4), 75 KCl, 75 NaCl, 0.5 TCEP, 20 maltose, 0.06% digitonin. MBP fusion tag was removed by incubation with Tobacco Etch Virus (TEV) protease overnight. NKCC1 was further separated with a Superose 6 column using buffer composed of (in mM) 20 HEPES (pH7.4), 100 NaCl, 50 KCl, 0.06% digitonin, and peak fractions corresponding to human NKCC1 were pooled and concentrated for cryo-EM analyses. For preparing NKCC1 bound with bumetanide, NKCC1 was purified with a Superose 6 column using a low [Cl$^-$] buffer composed of (in mM) 20 HEPES (pH 7.4), 50 Na$_2$SO$_4$, 25 K$_2$SO$_4$, 5 NaCl, 250 × 10$^{-3}$ bumetanide, 0.06% digitonin. The peak fractions corresponding to NKCC1 were supplemented with bumetanide to a final concentration of 500 μM and incubated for 2 h at room temperature prior to cryo-EM analyses. For preparation of NKCC1ii bound with furosemide, the same low [Cl$^-$] buffer supplemented with 250 μM furosemide was used during size exclusion chromatography. The peak fractions were pooled and supplemented with furosemide to a final concentration of 1 mM and incubated for 2 h at room temperature prior to cryo-EM analyses.

**Cl$^-$ influx assay in insect cells.** The NKCC1-mediated Cl$^-$ influx was measured using a membrane targeted yellow fluorescent protein (mbYFPQS) as a Cl$^-$ indicator[60]. Briefly, ~ 1.0 × 10$^5$ Sf9 insect cells were seeded per well in a poly-D-lysine treated, black-walled, clear-bottom 96-well plate and allowed to adhere for 30 min. Baculovirus co-expressing the wildtype human NKCC1 transporter (or mutants) and mbYFPQS was generated using pFastbac-Dual (Invitrogen) and added to the Sf9 culture; baculovirus only expressing mbYFPQS was included as a negative control in all experiments. 48–60 h post infection, the medium was replaced by 100 μl loading buffer (20 mM HEPES, 90 mM (NMDG)$_2$SO$_4$, pH7.4), and incubated for 2–3 h prior to assay. The loading buffer was exchanged to 100 μl assay buffer (20 mM HEPES, 120 mM NaCl, 60 mM KCl, pH7.4) to initiate NKCC1-mediated Cl$^-$ influx; 100 μM bumetanide was also added to half of the wells to inhibit NKCC1, validating that observed Cl$^-$ influx was mediated by NKCC1. For the MTSET inhibition assay, after conclusion of 2–3 h incubation in the loading buffer, 200 μM MTSET was added to the wells for an additional 15 min. The loading buffer was then exchanged to 100 μl assay buffer containing 200 μM MTSET to measure Cl$^-$ influx activity. Fluorescence intensity was measured on a BioTek Synergy Neo2 HTS Multi-Mode Microplate Reader (excitation/emission wavelengths are 485 nm/535 nm). The rates of Cl$^-$ transport were first calculated as the slopes of the fluorescent intensity change within the initial 20 s and then converted to mM/min unit based on a standard fluorescence *versus* [Cl$^-$] curve of mbYFPQS[61]. Unpaired Student's t test was

used to evaluate the significance of NKCC-mediated, bumetanide-sensitive transport activity.

**$^{86}$Rb$^+$ and Cl$^-$ influx studies in HEK cells.** Functional assays were carried out with human NKCC1 tagged with flag and YFP, and expressed in HEK293 cells; these were in a CRISPR-Cas9 NKCC1-knockout HEK line where noted. To assay NKCC1 activity in these cells we used both a $^{86}$Rb$^+$ influx assay and a Cl$^-$ influx assay, both of which have been described in detail[31,47,62]. In experiments where extracellular [Cl$^-$] was varied, we included rapidly reversible 200 μM furosemide during the preincubation to eliminate transport effects of NKCC; we also used high extracellular K$^+$ to maximize intracellular [Cl$^-$] and hypotonic 0 Cl$^-$ to rapidly reduce intracellular Cl$^-$ to 0 mM[47]. The $^{86}$Rb$^+$ influx assay[47] has the advantage of being able to determine activity over a wide range of intracellular Cl$^-$ concentrations ($^{86}$Rb is no longer commercially available in the US), whereas the Cl$^-$ assay allows more sensitive detection (<1% of wild type activity) using a Cl$^-$-sensing-YFP in Cl$^-$-depleted cells[31,62].

To evaluate the function of NKCC1iii, we transfected HEK293 cells with CFP, YFP-tagged NKCC1 constructs having CFP (cerulean) inserted at residue position 975, and YFP (mCitrine) at position 1117 in the C-terminus (NKCC$_{cy}$ and NKCC$_{iii-cy}$), analogous to the C0.49C-C0.80Y construct previously studied in shark NKCC1[58]. These experiments also tested an (A492E/L671C) mutant (NKCC$_{*ii-cy}$), which exhibited the same functional characteristics as NKCC$_{iii-cy}$. In FRET experiments carried out as described previously[58], the human construct (NKCC1$_{cy}$) behaved similarly to the reported shark construct sNKCC (C0.49C-C0.80Y), but the FRET$_{norm}$ decrease in response to calyculin A was only about 5% compared to the 11% change seen in sNKCC1. The YFP (mCitrine) in these constructs is not affected by [Cl$^-$] at physiological pH, so to measure [Cl$^-$] changes due to NKCC1 activity we used cells that were co-transfected with soluble Cl$^-$-sensing YFP – the signal from the soluble YFP is more than 10-fold that of the membrane-expressed FRET constructs.

**$^3$H-benzmetanide release from NKCC1 in shark rectal gland.** Release of $^3$H-benzmetanide (a loop diuretic analog with higher affinity and lower dissociation rate compared to bumetanide) from NKCC1 in the shark rectal gland was carried out using perfused glands of *Squalus acanthias* as previously described[37]. In these experiments, glands were stimulated with vasoactive intestinal peptide and perfused in normal Ringer's solution at 15 °C. In a brief (3 min) period at the beginning of the experiment K$^+$ or Cl$^-$ was substituted by another cation or anion as indicated in the figure -- for 1 min in the middle of the 3 min period, $^3$H-benzmetanide was included in the perfusate. During the rest of the subsequent perfusion, released $^3$H-benzmetanide was measured in the venous effluent as reported in the figures; 0.5 mM furosemide was included during this dissociation period to prevent rebinding of $^3$H-benzmetanide to NKCC1.

**Electron microscopy sample preparation and data collection.** For cryo-EM, 3.5 μl of NKCC1 sample at ~ 8–10 mg/ml, supplemented with/without 500 μM bumetanide, was applied to a glow-discharged Au 1.2/1.3 holey, 300 mesh gold grid and blotted for 1.5 s at 4 °C, 90% relative humidity on a Vitrobot Mark III (FEI) before being plunge-frozen in liquid ethane cooled by liquid nitrogen. Data were collected on a Krios (FEI) operating at 300 kV equipped with the K3 direct electron detector at the University of Utah and Pacific Northwest Cryo-EM Center (PNCC). Movies were recorded using SerialEM, with a defocus range between −1.0 and −3.5 μm. Specifically, movies were recorded in super-resolution counting mode at a physical pixel size of 1.06 Å. The data were collected at a dose rate of 1.175 e − / Å$^2$/frame with a total exposure of 40 frames, giving a total dose of 47 e − / /Å2.

**Image processing 3D reconstruction and model building.** Movie frames were aligned, dose weighted, and then summed into a single micrograph using MotionCor2[63]. CTF parameters for micrographs were determined using the program CTFFIND4[64]. Approximately 4000 particles were manually boxed out in cryoSPARC 3.0 to train a neuronal network model which was then used to extract particles from all micrographs using TOPAZ[65]. For the NKCC1iii/bumetanide dataset, a total of 504,481particles were extracted and then subjected to one round of 2D classification in cryoSPARC 3.0 software[66]. 'Junk' particles that were sorted into incoherent or poorly resolved classes were rejected from downstream analyses. The remaining 225,107 particles from well resolved 2D classes were pooled and were used to calculate four de novo models in cryoSPARC 3.0 without imposing any symmetry. The particles from the good ab initio class were then used to calculate a 3.1 Å map using non-uniform refinement in cryoSPARC 3.0. At this point, the 92,707 particles with orientation parameters were exported into RELION 3.0.7 for 3D classifications with extracellular domain masked out and with skip alignment enabled. The resulting 39,237 particles from one good 3D class were then subjected to non-uniform refinement in cryoSPARC 3.0, followed by CTF refinement and Bayesian polishing in RELION 3.0.7 software[67,68], eventually yielded a 2.9 Å map.

For the NKCC1ii/bumetanide dataset, a total of 1,033,074 particles were extracted and subjected to one round of 2D classification in cryoSPARC 3.0, resulting 369,313 good particles. Three ab initio models were calculated in cryoSPARC 3.0 and a single good class showing clear secondary structural features

and best map connectivity was selected and refined to 3.8 Å using non-uniform refinement in cryoSPARC 3.0. At this point, the 369,313 particles were exported into RELION 3.0.7 for several rounds of 3D classification using the 3.8 Å map as initial model. The 37,297 particles from the good 3D class were imported back to cryoSPARC 3.0 for non-uniform refinement, followed by CTF refinement and Bayesian polishing in RELION 3.0.7, and yielded a final map of 3.6 Å resolution.

For the NKCC1$_{iii}$/apo dataset, a total of 536,763 particles were extracted and subjected to one round of 2D classification in cryoSPARC 3.0, resulting in 320,030 good particles. Four ab initio models were calculated in cryoSPARC 3.0 and two good classes showing clear secondary structural features and best map connectivity were selected and refined to 3.7 Å using non-uniform refinement in cryoSPARC 3.0. At this point, the 320,030 particles were exported into RELION 3.0.7 for several rounds of 3D classification using the 3.7 Å map as initial model. The 75,001 particles from the good 3D class were imported back to cryoSPARC 3.0 for non-uniform refinement, followed by CTF refinement and Bayesian polishing in RELION 3.0.7, and yielded a final map of 3.3 Å resolution.

For the NKCC1$_{ii}$/furosemide dataset, a total of 1,521,495 particles were extracted from 4,338 micrographs and subjected to one round of 2D classification in RELION 4.0, resulting in 982,376 good particles. Four ab initio models were calculated in cryoSPARC 3.0 and one good class showing clear secondary structural features and best map connectivity were selected as initial model for 3D classification in RELION 4.0. The 85,360 particles from the good 3D class were imported back to cryoSPARC 3.0 for non-uniform refinement, followed by CTF refinement and Bayesian polishing in RELION 4.0, and yielded a final map of 3.4 Å resolution as reported by cryoSPARC.

The maps were locally sharpened in cryoSPARC 3.0 with an overall b factor of ~ −86 Å$^2$ (NKCC1$_{iii}$/bumetanide), 119 Å$^2$ (NKCC1$_{ii}$/bumetanide), or −160 Å$^2$ (NKCC1$_{iii}$/apo) for model building in Coot 0.8.9.3[69]. The human NKCC1 transmembrane structure (PDB: 6PZT) was dock into both maps and adjusted in Coot 0.8.9.3. The cytoplasmic C-terminal domain was built de novo into the NKCC1$_{iii}$/bumetanide map. Most of the human NKCC1 C-terminal domain can be unambiguously modeled, but the region encompassing residues 927-1021 was not defined. The models were refined in real space using PHENIX 1.18[70], and assessed in Molprobity[71] as shown in Supplementary Table 1. We did not build a final model for NKCC1$_{ii}$/furosemide since this structure is almost identical to NKCC$_{ii}$/bumetanide and because the map could not unambiguously define the pose for furosemide. FSC curves were then calculated between the refined model versus summed half maps generated in cryoSPARC 3.0, and resolution was reported as FSC = 0.5 (Supplementary Figs. 3–5). UCSF Chimera was used to visualize and segment density maps, and figures were generated using Chimera. jsPISA was used to calculate buried surface area[72]. Movies were created using the MORPHINATOR (http://morphinator.au.dk) tool developed by Jesper L. Karlsen, Aarhus University; the inward-facing model of human NKCC1 was generated from 6NPL.pdb with Modeller (https://salilab.org/modeller/) developed by Andrej Sali, UCSD.

**Molecular dynamic simulations**. NKCC1 monomer encompassing only the transmembrane region (282–753) was embedded in a POPC bilayer. The ionization state of titratable residues was determined using PropKa 3.0 software[73] assuming pH 7. The polypeptide chain was capped with acetyl and N-methyl groups at the N- and C- termini, respectively. The model included NKCC1 and bumetanide, together with 302 POPC molecules, 61 Cl$^-$, 33 K$^+$, 31 Na$^+$ ions (including the experimentally determined Cl$^-$ and K$^+$ ions) and ~22,000 water molecules (~115,000 atoms in total) in a simulation box of ~100 × 109 × 101 Å$^3$ size. Initial configurations were assembled using Packmol-Memgen[74].

MD simulations were performed with the GPU version of the PMEMD code of the AMBER package[75]. The ff14SB force field was used for the protein[76], Lipid17 for the POPC bilayer[77], TIP3P for water[78] and parameters from[79] for ions. The system was treated under periodic boundary conditions, using the particle mesh Ewald method to compute long-range electrostatics[80]. A 10 Å cutoff was used for the real part of the electrostatic and for van der Waals interactions. The SHAKE algorithm was used to constrain bonds involving hydrogen atoms[81], allowing an integration time step of 2 fs. Simulations were performed at constant temperature (310 K) and pressure (1 bar). The POPC bilayer and water solvent were allowed to equilibrate around the protein during 200 ns of MD simulations. After energy minimization, the system was gradually heated to 310 K, maintaining the protein backbone close to their positions in the cryo-EM structure by applying harmonic restraints. Then, about 1 μs of production MD were performed for each simulation of the Lys289Asn/Ala492Glu/Leu671Cys NKCC1 transporter.

**Human NKCC2 structure modeling and bumetanide docking**. The monomeric outward-open human NKCC2 structure was predicted by RoseTTAFold (https://robetta.bakerlab.org) using our outward-open human NKCC1 structure as a template. Bumetanide was docked into experimental human NKCC1 and predicted human NKCC2 structures, both of which harbor a K$^+$ ion essential for bumetanide binding, by the AutoDock Vina program[82]. The docking pose of bumetanide that ranks as the lowest free energy was chosen for further structural analyses.

**Reporting summary**. Further information on research design is available in the Nature Research Reporting Summary linked to this article.

## Data availability
The cryo-EM maps have been deposited in the Electron Microscopy Data Bank (EMDB) with the accession codes EMD-24807 (NKCC1$_{iii}$/bumetanide), EMD-24812 (NKCC1$_{ii}$/bumetanide), EMD-26588 (NKCC1$_{ii}$/furosemide), and EMD-24813 (NKCC1$_{iii}$/apo). The atomic coordinates for the corresponding maps have been deposited in the Protein Data Bank (PDB) with the accession codes 7S1X (NKCC1$_{iii}$/bumetanide), 7S1Y (NKCC1$_{ii}$/bumetanide), and 7S1Z (NKCC1$_{iii}$/apo). All other data and reagents that support the findings of this study are available from the corresponding authors upon request.

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

## Acknowledgements

This work was supported by the NIH grant 1R01 DK128592. E.C. is a Pew Scholar supported by the Pew Charitable Foundation. We thank Anita Orendt, Irvin Allen, Martin Cuma, and other staff members at the Utah Center for High Performance Computing for computational support. We are grateful to David Timm and David Belnap for data collection at the Electron Microscope Core at the University of Utah. The Electron Microscope Core at the University of Utah was supported by a grant from the Beckman Foundation. A portion of this research was supported by NIH grant U24GM129547 and performed at the PNCC at OHSU and accessed through EMSL (grid.436923.9), a DOE Office of Science User Facility sponsored by the Office of Biological and Environmental Research. Some of this work was performed at the National Center for CryoEM Access and Training (NCCAT) and the Simons Electron Microscopy Center located at the New York Structural Biology Center, supported by the NIH Common Fund Transformative High Resolution Cryo-Electron Microscopy program (U24 GM129539), and by grants from the Simons Foundation (SF349247) and NY State Assembly. We thank Claudia Lopez, Harry Scott, Janette Myers, Drew Gingerich, Ed Eng, Elina Kopylov, and other staff members at the PNCC and NCCAT for data collection and technical support.

## Author contributions

Conceptualization: Y.Z. and E.C. designed cryo-EM studies and $Cl^-$ influx assays, and B.F. designed constructs and guided functional studies, P.V., L.C., and M.D.V. designed MD simulations; Investigation: Y.Z. carried out cryo-EM experiments and $Cl^-$ influx assays in insect cells, and B.F. and K.R. performed ion flux assays in HEK293 cells and bumetanide-binding studies, P.V. conducted MD simulations; Writing: E.C., Y.Z., and B.F. wrote the manuscript with contribution of all other authors.

## Competing interests

Laura Cancedda and Marco De Vivo are founders of the IAMA Therapeutics. All other authors declare no competing interests.
