## [Peer Review File · Nature Communications]

Structural Basis for Inhibition of the Cation-chloride Cotransporter NKCC1 by the Diuretic Drug BumetanideREVIEWER COMMENTS

Reviewer #1 (Remarks to the Author):

Zhao et al report structures of NKCC1 in complex with the loop diuretics bumetanide in the outward-facing conformation. They also present a new NKCC1 dimer with separated TM domains and extensive TM and C-terminal domain interactions. This work greatly improves our understanding of NKCC1 structures and mechanisms. The structure determinations are solid and supported by activity assays and MD simulations.

Major concerns

1. With the presence of bumetanide, the authors were able to capture the outward-facing conformation of NKCC1. The density of bumetanide is strong and clear. However, it should be cautious to interpret the ligand-stabilized outward-facing conformation, which is potentially an artifact induced by the binding of the ligand. The authors should perform functional assays and MD simulations to confirm the existence of this conformation without ligand.
2. Similarly, the functional validation of the NKCC1iii/bumetanide structure with separated two TMD should also be performed. Does this state really exist in vivo? Or an artifact only observed in vitro? Previous FRET assays showed the dynamics of the CTD when the transporter works. A similar FRET assay should help to address this question.
3. In the outward-facing conformation, the authors show the density of Cl₂. Is this Cl⁻ site solvent-accessible from the extracellular side? Similarly, is the Na site accessible from the extracellular side? The interpretation of the Na⁺ and Cl₂ sites should be careful. Whether these two ions identified in the zebrafish NKCC1 are the substrates are questionable, because 1) Na⁺ site is away from the canonical substrate transport pathway in APC family; and 2) Cl₂ is also observed in KCCs not proposed as a non-transport ion.

Reviewer #2 (Remarks to the Author):

In this paper, Zhao et al. investigate the basis for inhibition of human NKCC1 by the loop diuretic bumetanide. They determine cryo-EM structures of NKCC1 in an outward-open state (for the first time, by introducing mutations that favor that state) and identify a bumetanide binding site in the outer ion transport pathway. Two additional new structural features are identified and analyzed: a N-terminal region that is involved in regulation of ion transport by phosphorylation and dissociation of an intersubunit transmembrane interface within a homodimer. This is a very interesting and well performed study that provides novel insights into a physiologically important transporter and its interaction with a clinically important drug. It will be of broad interest. I have several comments that should be addressed prior to publication.

1. I do not understand how Supp. Fig. 1 shows A492E has constitutive activity. A492E appears to show the same influx as WT after preincubation in high Cl and reduced activity in low Cl, which accounts for the larger ratio in 140/0 Cl. Does this suggest A492E rather promotes an outward open state at rest?

2. The impact of mutations introduced in the NKCC1ii/NKCC1iii constructs on conformation and activity should be discussed.

3. Perhaps it could be made clear that major differences exist in the three structures (in transmembrane domain association, association of the N-terminal and C-terminal regions, and relative order of the C-terminal region) in the first paragraph of the results section and that these will be discussed later.

4. It would be useful to include a comparison of the bumetanide binding site in apo and bound structures to illustrate how drug binding involves the Cl⁻ site¹.

5. One prediction of the model for association of N- and C- terminal regions is that strengthening this interaction (e.g. through phosphorylation at T217) increases transport activity. Could the authors design mutations to strengthen the interaction between N-terminal region and C-terminal domain in the absence of phosphorylation? Increased activity in these mutants (or rescue of a T217A mutant) would provide more convincing evidence for the model than the mutations in Fig. 2 that disrupt the interaction and reduce activity.

6. The authors should elaborate on why the NKCC1iii/bumetanide adopts a conformation with separated transmembrane domains while the NKCC1iii/apo, NKCC1ii/bumetanide and previous NKCC1 structures do not. What results in the different positioning of the TM12-C-terminal domain linker in the NKCC1iii/bumetanide structures?

7. Does association of N-term and C-term regions require TM disengagement?

8. The authors should consider including (in the discussion) reference to previous literature suggesting changes in oligomeric state and structural evidence of different oligomeric states in KCC transporters in addition to the discussion on changes in NKCC1 dimer arrangement (Blaesse et al. J. Neurosci. 2006; Puskarjov et al. J. Neurosci. 2012; Watanabe et al. JBC 2009; Liu et al. Science 2019; Reid et al. eLife 2020).

We thank the reviewers for their constructive critiques of our initial manuscript. In this resubmission, we have addressed reviewers concerns with additional experiments. In summary, our major advances are: 1) We have performed further molecular dynamic (MD) simulations to validate the physiological relevance of the outward open state we captured using bumetanide. These new simulations showed that after bumetanide is removed the outward open conformation seen in our NKCC1/bumetanide structures remains stable throughout the duration of the simulations. 2) We have carried out accessibility of substituted cysteine assays to ascertain physiological relevance of the extracellular vestibule revealed by our outward open NKCC1 structures. Extending previous reports, we found that substitution of three residues lining the extracellular ion translocation path does not affect NKCC1 activity, but renders the transporter sensitive to inhibition by a cysteine reactive compound MTSET. These new data strong support the extracellular ion translocation path revealed by our outward open NKCC1 structures. 3) We have obtained a 3.4 angstrom NKCC1/furosemide structure that supports our hypothesis that the extracellular ion translocation path of NKCCs represents a general binding site for loop diuretics. 4) We used a more sensitive ion flux assay to show that the NKCC1_{iii} construct retains reduced but significant transport activity compared to wild type, confirming functionality of the constructs used for cryo-EM studies. 5) We carried out FRET experiments that implicate the C-terminal domain in activation of the NKCC1_{iii} construct. Our point-to-point responses to the reviewers are outlined below.

Reponse to Reviewer #1:

Major concerns

1. With the presence of bumetanide, the authors were able to capture the outward-facing conformation of NKCC1. The density of bumetanide is strong and clear. However, it should be cautious to interpret the ligand-stabilized outward-facing conformation, which is potentially an artifact induced by the binding of the ligand. The authors should perform functional assays and MD simulations to confirm the existence of this conformation without ligand.

We agree there is a possibility that bumetanide could induce a conformation dissimilar from transporting states. As suggested by the reviewer, we have now carried out further MD simulations that validate the stability of the outward open state seen in our structures after bumetanide is removed (**Supplementary Fig. 13c**). In a complementary approach to further validate physiological relevance of our NKCC1/bumetanide structures, we measured accessibility of substituted cysteines that line the extracellular ion translocation path seen in our outward open structures. We found that, in contrast to the wildtype NKCC1 transporter, three such cysteine substitution mutants show sensitivity to inhibition by a cysteine reactive reagent MTSET (**Supplementary Fig. 20**). Together, these new data support the idea that bumetanide selectively binds to and stabilizes a pre-existing and physiologically relevant outward open state of NKCC1.

2. Similarly, the functional validation of the NKCC1_{iii}/bumetanide structure with separated two TMD should also be performed. Does this state really exist in vivo? Or an

artifact only observed in vitro? Previous FRET assays showed the dynamics of the CTD when the transporter works. A similar FRET assay should help to address this question.

We again agree with the reviewer that the physiological importance of our TMD-separated dimer remains to be explored. MD simulations of the NKCC_{iii}/bumetanide dimer are computationally prohibitive at this stage. We have however further tested the NKCC1_{iii} construct and found that it retains a low level of activity, and that bumetanide inhibits its activity with higher affinity than for the wild type. As suggested by the reviewer, we performed FRET experiments using NKCC1_{iii} tagged with a FRET pair in the C-terminal domain of the transporter. We found a similar, albeit smaller, response compared to tagged wild type hNKCC1, indicating that this construct is regulated in manner similar to the wild type (**Supplementary Fig. 4**). These experiments confirm the functionality of our cryo-EM construct, but fall short of answering the very difficult question of whether the TMD-separated dimer with bound bumetanide exists in vivo. We hope that our future single-molecule FRET experiments mentioned at the end of the Discussion section will be able to provide an answer to this and similar questions.

3. In the outward-facing conformation, the authors show the density of Cl₂. Is this Cl-site solvent-accessible from the extracellular side? Similarly, is the Na site accessible from the extracellular side? The interpretation of the Na⁺ and Cl₂ sites should be careful. Whether these two ions identified in the zebrafish NKCC1 are the substrates are questionable, because 1) Na⁺ site is away from the canonical substrate transport pathway in APC family; and 2) Cl₂ is also observed in KCCs not proposed as a non-transport ion.

When bumetanide is removed from our outward-facing NKCC1/bumetanide structures, the K⁺ ion is fully accessible from the extracellular space, whereas Cl⁻ at site 2 and Na⁺ ions (both tentatively assigned) appear to be inaccessible from both the extracellular and intracellular sides (see the Figure on the right). Thus we agree with the reviewer that experimental evidence for assigning Cl⁻ site 2 and Na⁺ site remain lacking, and in the updated main text we now refer to these sites as tentative.

Response to Reviewer #2:

1. I do not understand how Supp. Fig. 1 shows A492E has constitutive activity. A492E appears to show the same influx as WT after preincubation in high Cl and reduced activity in low Cl, which accounts for the larger ratio in 140/0 Cl. Does this suggest A492E rather promotes an outward open state at rest?

The reviewer is correct, we were over-enthusiastic in describing A492E as constitutively active, we have now restricted that term to describing L671C. We were looking for mutations that would increase the outward-open frequency – as the reviewer suggests it is possible that this occurs without involving the activation machinery, but it is also possible that it increases the low but finite occupancy of activated states “at rest”; currently we cannot distinguish.

2. The impact of mutations introduced in the NKCC1ⁱⁱ/NKCC1ⁱⁱⁱ constructs on conformation and activity should be discussed.

As outlined above, we have now studied the functionality of NKCC1ⁱⁱⁱ using a more sensitive ion flux assay in HEK293 cells and a FRET approach (**Supplementary Figs. 3 and 4**). Together, these new data show that NKCC1ⁱⁱⁱ does exhibit bumetanide-sensitive ion transport activity and that activation of our cryo-EM construct involves movement of the C-terminal domain similar to the wildtype NKCC1 transporter. We point out the impact of the mutations on activity, and suggest that this may occur by biasing towards the outward-open state and reducing turnover (supported by higher affinity for bumetanide).

3. Perhaps it could be made clear that major differences exist in the three structures (in transmembrane domain association, association of the N-terminal and C-terminal regions, and relative order of the C-terminal region) in the first paragraph of the results section and that these will be discussed later.

Yes, this is now addressed.

4. It would be useful to include a comparison of the bumetanide binding site in apo and bound structures to illustrate how drug binding involves the Cl⁻ site1.

This is now included in the updated Supplementary Fig. S12b. This new Figure illustrates that the carboxyl group of bumetanide clashes with Cl⁻ at site 1, showing that bumetanide and the Cl⁻ ion in site 1 are mutually exclusive.

5. One prediction of the model for association of N- and C- terminal regions is that strengthening this interaction (e.g. through phosphorylation at T217) increases transport activity. Could the authors design mutations to strengthen the interaction between N-terminal region and C-terminal domain in the absence of phosphorylation? Increased activity in these mutants (or rescue of a T217A mutant) would provide more convincing evidence for the model than the mutations in Fig. 2 that disrupt the interaction and reduce activity.

We completely agree with the reviewer on this point, a gain-of-function NKCC1 mutant would provide valuable insight into phosphoregulation of NKCCs. We have spent significant effort in search of such a mutant without success, which are now

summarized in **Supplementary Fig. 18**. We showed that T217A and T230A lose transport activity, validating the roles of these two key phosphorylation sites. However, mutations, including two phosphomimetic mutants (T217E and T230E) that we hoped would enhance association between N- and C-terminal domains, all exhibit decreased activity; we also searched for a second mutation that could rescue the loss-of-function phenotype of T217A without success. These results add to a number of similar previously published failed attempts. One possible explanation for the difficulty in inducing gain-of-function phenotype is that the inactive state may be required for correct cellular processing of NKCCs.

A gain-of-function mutation is also theoretically possible by improving the “design” of the region downstream of the phosphorylation site identified here as the N-C interacting region. Unfortunately, in the absence of what would be extensive modeling, we do not have insight as to mutations that would improve upon the structure that already exists.

6. The authors should elaborate on why the NKCC1ⁱⁱⁱ/bumetanide adopts a conformation with separated transmembrane domains while the NKCC1ⁱⁱⁱ/apo, NKCC1ⁱⁱ/bumetanide and previous NKCC1 structures do not. What results in the different positioning of the TM12-C-terminal domain linker in the NKCC1ⁱⁱⁱ/bumetanide structures?

This is also an important question for us. When comparing zebrafish NKCC1, our human NKCC1ⁱⁱⁱ/bumetanide, and alphafold-predicted human NKCC1 structures, we noticed that the link region engages in more extensive interactions with the TM domain in our NKCC1ⁱⁱⁱ/bumetanide structure than in other two NKCC1 structures (see the updated **Figure 5b**). This suggests that conformational malleability of this linker region could accommodate separation of the two transmembrane units.

7. Does association of N-term and C-term regions require TM disengagement?

We do not have the answer for this intriguing question. Most of the N-terminal region is flexible and lacks predicted secondary structure, while the C-terminal domain could assume a range of conformations -- these observations argue against the hypothesis that N- and C-domain association directly exert force on the transmembrane units. On the other hand, the stable association between the N- and C-terminal domains of NKCC1 apparently constrains cytosolic domains of human NKCC1 in a way not observed in our two other human NKCC1 structures. Stabilization through N- and C-domain interactions coincides with a newly formed C-domain and TM interface and with separation of the two TM units, but at this point we cannot say that the N-C domain association *requires* TM disengagement.

8. The authors should consider including (in the discussion) reference to previous literature suggesting changes in oligomeric state and structural evidence of different oligomeric states in KCC transporters in addition to the discussion on changes in NKCC1 dimer arrangement (Blaesse et al. J. Neurosci. 2006; Puskarjov et al. J.

Neurosci. 2012; Watanabe et al. JBC 2009; Liu et al. Science 2019; Reid et al. eLife 2020).

We have now included these references (except Puskarjov et al.) in the discussion. It is outside of the scope of this manuscript, but we find literature reports of NKCCs and KCCs dimer association/dissociation based simply on SDS gels to be hard to interpret at best. The critically important point is that sodium dodecyl sulfate is a strong ionic detergent that disrupts subunit associations, so that the use of SDS gel electrophoresis to determine native oligomerization state is an inherently misleading exercise. In our own experiments, we (Forbush) find that freshly solubilized NKCC1 (which we know from many types of experiments is primarily a dimer in the membrane) runs as a monomer on SDS gels, and we have found that the dimer seen on gels is an artifact of cross-linking during post-solubilization processing. NKCC1 is exquisitely prone to covalent dimerization in the test tube under a wide variety of conditions, including those that are used in biotin-based precipitation to determine cell-surface labelling. In the simplest case, this leads to the erroneous conclusion that (a) total cell NKCC1 includes a lot of monomer (the monomer is created by SDS dissociation of the native dimer), and the correct conclusion (for the wrong reason) that (b) cell-surface NKCC1 is a dimer (it is a dimer in the membrane, and much of that dimer is covalently crosslinked during the precipitation procedures, so it runs on an SDS gel as a dimer as well). Notwithstanding the recent demonstration of Reed et al. that KCC4 can be reconstituted in nanodiscs as a monomer, we know of no convincing evidence that NKCC1 is ever anything but a dimer in a native membrane, and we suspect the same may be true for KCCs.

REVIEWERS' COMMENTS

Reviewer #1 (Remarks to the Author):

The authors have addressed all my major concerns and I have no further question. This manuscript is of high quality and provides solid structural and biochemical data for the diuretic drug recognition by NKCC1, a very important step for the development of new drugs targeting NKCCs. I recommend it to be published in NC as soon as possible.

Reviewer #2 (Remarks to the Author):

The authors have nicely addressed all of my concerns in this revised manuscript and response. I have no further concerns.